# Spatial Equity of Multilevel Healthcare in the Metropolis of Chengdu, China: A New Assessment Approach

**DOI:** 10.3390/ijerph16030493

**Published:** 2019-02-10

**Authors:** Shaoyao Zhang, Xueqian Song, Yongping Wei, Wei Deng

**Affiliations:** 1Research Center for Mountain Development, Institute of Mountain Hazards and Environment, Chinese Academy of Sciences, Chengdu 610041, China; zhangsyxs@163.com (S.Z.); dengwei@imde.ac.cn (W.D.); 2School of Resources and Environment, University of Chinese Academy of Sciences, Beijing 100049, China; 3School of Management, Chengdu University of Information Technology, Chengdu 610225, China; 4School of Earth and Environmental Sciences, the University of Queensland, Brisbane 4067, Australia; yongping.wei@uq.edu.au

**Keywords:** multilevel healthcare, spatial equity, spatial accessibility, 2R grid-to-level (2R-GTL)

## Abstract

The spatial equity of the healthcare system is an important factor in assessing how the different medical service demands of residents are met by different levels of medical institutions. However, previous studies have not paid sufficient attention to multilevel healthcare accessibility based on both the divergence of hierarchical healthcare supplies and variations in residents’ behavioral preferences for different types of healthcare. This study aims to propose a demand-driven “2R grid-to-level” (2R-GTL) method of analyzing the spatial equity in access to a multilevel healthcare system in Chengdu. Gridded populations, real-time travel distances and residents’ spatial behavioral preferences were used to generate a dynamic and accurate healthcare accessibility assessment. The results indicate that significant differences exist in the spatial accessibility to different levels of healthcare. Approximately 90% of the total population living in 57% of the total area in the city can access all three levels of healthcare within an acceptable travel distance, whereas multilevel healthcare shortage zones cover 42% of the total area and 12% of the population. A lack of primary healthcare is the most serious problem in these healthcare shortage zones. These results support the systematic monitoring of multilevel healthcare accessibility by decision-makers. The method proposed in this research could be improved by introducing nonspatial factors, private healthcare providers and other cultural contexts and time periods.

## 1. Introduction

The unbalanced distribution of healthcare resources across space and healthcare levels is an important reason for disordered medical treatment [1,2]. The Hierarchical Diagnosis and Treatment (HTD) reform, which was implemented in China in 2015, aimed to improve the efficiency and equity of multilevel healthcare. Recent decades have witnessed considerable developments in urban spaces, and the methods of allocating healthcare resources must be considered based on the changing population configuration and the development of transportation in a metropolis. In these contexts, dynamic spatial equity in access to a multilevel healthcare system in a metropolis should be a focal point of scholarly attention. 

Spatial equity in healthcare concerns the perceived lack of fairness in the well-being of individuals and societies across space. Inequitable levels of access have important effects on health outcomes [3]. Research that investigates geographic accessibility provides a basis for optimizing the allocation of healthcare [4,5]. Geographic accessibility can be defined as the ease with which the residents in a given area can reach a particular service or facility [6], and it includes the availability of and proximity to healthcare [7]. Availability indicates that the services and facilities can meet the residents’ needs, and proximity indicates that the residents can visit the services and facilities based on their own capabilities [8]. Healthcare geographic accessibility can be influenced by many factors, such as the availability of healthcare sites in the area (supply), the population of the area (demand) and the geographic barriers between supply and demand [9]. 

The gravity model and the two-step floating catchment area (2SFCA) method are the mainstream models used to assess the geographic accessibility to healthcare [10,11]. The gravity model is used to evaluate healthcare accessibility by analyzing variations in the ratio of healthcare suppliers to the population demand based on distance decay. Although the methods used to calculate the distance, population, and capacity of healthcare facilities have consistently improved, the results are not accurate due to uncertainty in the impedance coefficient [12]. The 2SFCA method is applied to evaluate healthcare accessibility by defining the range of the catchment area threshold for hospitalization and the scope of healthcare services. The basic model has undergone various incremental enhancements, including the incorporation of variable catchment sizes, distance decay effects, age-adjusted healthcare demand and multiple transport modes [5,13,14,15]. However, these measures of (equity in) accessibility in empirical health studies still tend to be conceptualized predominantly in static spatial terms. Most studies analyze the spatial distribution of people demanding healthcare based on statistical data; catchment size is often defined arbitrarily; and residents’ healthcare-seeking behaviors, such as their transportation choices and the speeds and routes, are assumed to be constrained under certain conditions [6]. Healthcare accessibility assessments are not accurate because of a lack of detailed estimates of the heterogeneity of the population distribution and residents’ dynamic spatial behavioral patterns [16]. Therefore, the need for more spatially disaggregated, individualized and temporally aware accessibility metrics has been acknowledged [17,18,19].

The accessibility to different levels of healthcare services, ranging from large comprehensive medical institutions to primary healthcare services, has been assessed [20,21,22]. Scholars have also paid attention to the different capacities of different medical institutions and have evaluated the number of doctors and hospital beds [23]. Healthcare institutions have been classified in terms of scale and service capacities, and different weights have been assigned to these institutions to evaluate the impacts of different service capacities on the healthcare service spatial distribution [24,25]. These studies provide significant insights into the spatial accessibility to healthcare from the perspective of the healthcare supply; however, residents’ multilevel healthcare demands have not been sufficiently considered. Current studies suggest that the upper levels of medical treatment can replace the lower levels [16]. In practice, however, residents should choose different levels and types of medical institutions according to their different needs, and their spatial behavioral preferences for seeking different levels of healthcare are quite different. 

In the current healthcare system in China, most of patients’ access to healthcare services is not organized according to a gatekeeping system and a two-directional referral network. Therefore, upper-level hospitals are always overcrowded, while lower-level health centers have fewer patients. This situation increases medical costs, wastes healthcare resources and lowers healthcare efficiency [1,2]. Given this background, the HDT reform is considered an effective solution to defuse the antagonistic relationship between the use of healthcare resources and residents’ medical treatment demands. This reform emphasizes that residents’ different medical service demands correspond to the different levels of medical institutions. If a lack of medical services occurs at a certain level, then assessments of the accessibility to multilevel healthcare in that area should be lowered. However, previous studies have not paid sufficient attention to multilevel healthcare accessibility based on both the divergence of hierarchical healthcare supplies and variations in residents’ behavioral preferences for different types of healthcare. 

This paper aims to develop an approach to provide demand-driven assessments of the spatial equity in multilevel healthcare accessibility in Chengdu, a metropolis in China. Population gridding based on night-light data was used to obtain spatially disaggregated living configurations, and the residents’ behavioral preferences obtained from a questionnaire survey and real-time travel distances provided by an electronic map navigation service were employed to obtain dynamic, accurate and individualized transportation patterns. Rather than each hospital, each level of healthcare according to residents’ different healthcare needs was utilized as the research object to evaluate the macro and systematic healthcare spatial distribution in the metropolis. 

The remainder of the paper is organized as follows: Section 2 presents a brief introduction of the study area, the study method and the data used in this analysis. Section 3 reports the empirical results, and Section 4 make a comprehensive discussion of the results and highlights the main conclusions drawn from this study in Section 5. 

## 2. Methods

### 2.1. Case study Description

Chengdu is among the eight national central cities in Southwest China and the capital of Sichuan Province, and it represents the economic, scientific, and educational center of the western region of China. There are 11 urban districts, five prefecture-level cities, five counties, and 375 township communities in Chengdu, covering 14,335 km^2^. Plains, hills and mountain areas account for 40.1%, 27.6%, and 32.3% of the total area, respectively. Longquan Mountain and Longmeng Mountain are east and west of the city. Longmeng Mountain is rugged and remote with many forests and parks, and Longquan Mountain is connected to the central city through several highways and railways. The plains region has an average altitude of 500 m and is located in the middle of the city, which is the major area with regard to the population distribution and economic development. The permanent resident population in Chengdu reached 15.7 million by the end of 2015, and the city has a GDP of 1120.2 billion yuan and an urbanization rate of 71.5%. With a large population, varied topography, rapid changing urban space, and relatively substantial healthcare resources, Chengdu represents an excellent study area for examining equalities in spatial access to multilevel healthcare services.

### 2.2. Methods

Figure 1 provides an overview of the “2R grid-to-level” (2R-GTL) approach, which is represented as a dynamic demand-driven multilevel healthcare spatial equity evaluation. The term “grid” refers to 1-km^2^ grids, which are both the units of spatialization of the permanent population and the origin of the spatial accessibility calculation; the term “level” represents the three levels of medical institutions in a multilevel healthcare system according to China’s HDT reform; and the term “2R” refers to real-time traffic distance and residents’ behavioral preferences for different levels of healthcare. The shortest travel time from each grid to different levels of medical institutions was calculated based on the real-time travel distance according to the residents’ spatial behavioral preferences. Thus, the spatial accessibility of each level of healthcare was distinguished and overlapped. Finally, the area and population ratio of different types of multilevel healthcare access zones were identified to analyze the spatial equity of the multilevel healthcare system.

#### 2.2.1. Gridding of the Permanent Resident Population

The spatialization of the permanent resident population was based on the townships’ population data and night-light images. The monthly average night-light value of Chengdu in 2015 was coordinated with the population data for the township communities at a 1×1 km resolution. In addition, the noise was removed using 2013 Chengdu nightly average stability data (DMSP-OLS) set as a mask [26]. The permanent resident population was allocated to 1-km^2^ grids according to the ratio of the light value per cell to the overall strength of the effective. The formula is as follows:(1)$$G_{ij}=\frac{D_{ij}}{\sum_{i=1}^{n}D_{ij}}\times T_{j}$$
where $G_{ij}$ is the permanent resident population in cell i in township j; $D_{ij}$ is the valid value of cell i in township j; $T_{j}$ is the permanent resident population in township j; and n is the cell number of effective night-light data in township j.

The permanent resident population in the grid cells was compared to the statistical population data by relative error correction and accuracy testing. The formula is as follows [27]:(2)$$\delta =\frac{G_{p}-G_{s}}{G_{s}}\times 100\%$$
where δ is the relative error correction; $G_{p}$ is the sum of the spatial gridding of the permanent resident population in 2015; and $G_{s}$ is the sum of the statistical data of the permanent resident population in 2015.

The population value in each grid unit was adjusted according to the correction coefficient of each township using the following formula:(3)$$G_{ij}{}^{/}=G_{ij}\times \left(\frac{T_{j}}{G_{j}}\right),\;G_{j}=\sum_{i=1}^{n}D_{ij}$$
where $G_{ij}{}^{/}$ is the corrected value of the permanent resident population of cell *i* in township *j*; and $G_{j}$ is the sum of the spatial gridding of the permanent resident population in township *j*.

#### 2.2.2. Calculation of Healthcare Spatial Accessibility

Travel distance was calculated by the 2R (residents’ treatment preferences and real-time traffic data) method. Residents’ traffic choices, the ordinary acceptable travel time and the maximum travel time in seeking different levels of medical institutions were obtained from the residents’ spatial behavioral preferences survey. The traffic distances from each grid center to a certain level of healthcare were calculated using the real-time traffic data provided by the Baidu Navigation Service. The calculation was executed from 9:00–12:00 a.m. on weekdays to avoid rush hours, holidays, and other special situations.

Grid-to-level healthcare accessibility was determined based on 1 × 1 km grid nets of Chengdu and by calculating the travel time from each grid center to each level of the healthcare system. To reduce the time for the calculation, we computed the geographic distance from one grid center to each hospital at a specific level of healthcare and ranked the spatial distance to identify the five nearest hospitals. Then, the travel time from the grid centers to these five hospitals was calculated, and the shortest time was assigned to the grids (Figure 2). Finally, the shortest travel time from each grid to each level of healthcare was obtained. If the shortest travel time from one grid to a specific level of healthcare was less than or equal to the ordinary acceptable travel time, then this grid was defined as in the core service area (CSA) for this level. However, if the shortest travel time from one grid was more than the ordinary acceptable time and less than or equal to the maximum travel time, then the grid was defined as an effective service area (ESA). Regions outside the ESA were defined as healthcare shortage areas (HSAs). All calculations were executed by coding in Python 3.6 (manufacturer, city, state abbreviation if USA, country Guido van Rossum, Netherlands). 

### 2.3. Data Resources and Processing

The data used in this study can be divided into the following five categories:

*Medical institution data*. Based on the HDT reform, the multilevel public healthcare service system is classified into three levels. The definition and function of each level are shown in Table 1.

Data regarding the numbers and locations of the city-level and county-level hospitals were obtained from the Sichuan Provincial Health and Family Planning Statistical Yearbook 2015, and the data on the community-level health centers were obtained from point of interest (POI) data on the Baidu map (https://map.baidu.com/). There are 629 public healthcare institutions in Chengdu, including 15 city-level hospitals, 64 county-level hospitals and 550 community-level health centers. The medical institution address data are transformed to geographic spatial data based on the Baidu geocoding API (http://lbsyun.baidu.com/index.php?title=webapi/guide/webservice-geocoding) (Figure 3).

*Permanent resident population data*. Permanent resident population data for the 375 township communities were obtained from the yearbooks of all districts and counties in Chengdu and the Statistical Yearbooks of China’s Districts and Counties 2015 (township volumes). 

*Night-light image data*. The night-light image data were obtained from the NOAA NPP/VIIRS night light dataset (https://ngdc.noaa.gov/eog/viirs/download_dnb_composites.html). This dataset is obtained with no cloud images from the Suomi NPP satellite, which tests six regions of the spectrum, and the subpixel infrared sensor; this night-light image has a spatial resolution that reaches 15″ (approximately 450 m) and was updated until 2015.

*Residents’ behavioral preferences data*. To understand the residents’ spatial behavioral preferences in seeking multilevel healthcare, the research team conducted a questionnaire survey via face-to-face interviews at the sample’s city-level and county-level hospitals and community-level health centers in July 2015. The sample healthcare institutions were selected from different spatial circular layers of the city. Both outpatients and inpatients or their accompanying family members and friends were interviewed in different hospital departments. In total, 1079 valid questionnaires were obtained. 

*Real-time traffic data*. Real-time road traffic status data were obtained from the Baidu route direction service API v2.0 (http://lbsyun.baidu.com/index.php?title=webapi/guide/webservice-geocoding). Baidu map is the most popular electronic map service provider in China, and its navigation services provide route planning, mileage, and travel time, while the trip mode includes driving, public transportation, taxi, cycling and walking.

## 3. Results

### 3.1. Gridded Permanent Resident Population

Figure 4 illustrates the results of the permanent residents’ population gridding process. Figure 4a presents the population density distribution based on the population census statistics for each township community at the end of 2015, and it shows the population clustering trend and the spatial structure of a circular layer in Chengdu. The spatial structure has formed over more than 3000 years of human history, and it significantly influences the population distribution. The population density is the highest in the city center and decreases in the peripheral areas to the lowest populations in the Longquan and Longmeng Mountains. In addition, there are population concentrations in some suburb satellite cities, such as Wenjiang, Pidu and Dujiangyan. This centripetal distribution pattern indicates that the population has settled in areas with larger urban agglomerations, while some residents have scattered into the remote mountainous areas east and west of the city. 

To obtain a more disaggregated population distribution pattern, the NPP-VIIRS night-light data from 2015 (Figure 4b) were applied to analyze the gridded population distribution. As shown in Figure 4c, the total population in Chengdu is 15.9 million, which is 196.7 thousand higher than that reported in the statistical data. This difference may have been caused by double counting during the gridding process because some grids cross more than one administrative area. The relative error in the population spatialization in this research was 1.25% based on formula (2), which is superior to that reported in similar studies [27]. To further test the precision of the gridded population distribution, a correlation analysis between the gridded population and the census population of each township was performed. As shown in Figure 4d, a significant linear correlation with a high degree of fit (R^2^ = 0.9859) is observed between these two series of data. Therefore, the method of spatialization of the population fits the actual population distribution on a 1-km grid scale; hence, the healthcare spatial accessibility can be evaluated accurately. In addition, Figure 4a only shows the population distribution differences among the administrative areas, whereas Figure 4c shows the gridded population data and illustrates the distribution and aggregation of the population within administrative areas. Using the gridded population method, the global spatial pattern and heterogeneous spatial distribution of the population can be simultaneously presented. Compared to Figure 4a, Figure 4c shows that there are many suburban centers with high population density, and the centripetal population distribution pattern is more obvious.

### 3.2. Multilevel Healthcare Spatial Accessibility

#### 3.2.1. Residents’ Spatial Behavioral Preferences 

The results of the investigation indicate that a significant difference exists in residents’ traffic choices and distance tolerance when visiting different levels of medical institutions (Table 2). Moreover, the different types of access areas are defined according to these spatial behavioral preferences. 

#### 3.2.2. Multilevel Healthcare Spatial Accessibility

Figure 5 presents the spatial accessibility results for each level of healthcare and the overlapped multilevel healthcare access zones. The different-colored grids in the first three maps indicate the different travel times required to visit each level of healthcare and the different catchment sizes of each level of healthcare. The spatial equity in access to multilevel healthcare was analyzed by identifying the areas and population proportions of different types of multilevel healthcare access zones.

*City-level healthcare accessibility*. As shown in Figure 5a, 13 out of 15 city-level hospitals were in the city center. Moreover, the other two hospitals were in Wenjiang District (west of the city), which greatly enhanced the spatial accessibility of this region. Only one city-level hospital was in Longquan District (east of the city). The city center and most suburban areas are in the city-level hospital ESA, which covers 96.67% of the total area in the city. In addition, the CSA covers 51.69% of the total area. The city-level HSAs are mainly located in the west in the sparsely populated Longmen Mountain area. Regarding the eastern area, the barrier caused by Longquan Mountain is not significant because several highways and roads enhance traffic accessibility. However, a few remote communities in this area are still in the HSA.

*County-level accessibility*. At least one public comprehensive hospital was allocated to each district or county; thus, the equilibrium of the county-level hospital distribution was relatively higher than that of the city-level hospitals. However, according to the travel distances based on the residents’ behavioral preferences, the accessibility of county-level hospitals was significantly unequal (Figure 5b). In terms of a global spatial pattern, the ESA of county-level healthcare covered 88.26% of the total area, and the CSA covered only 61.92%. The level of accessibility is relatively low in the western and northern mountainous areas. Some settlements are located in county-level HSAs. More than 2 hours are required for residents living in the Xiling Snow Mountain scenic area to visit the nearest county-level hospital. Moreover, in the east, the spatial disparity in the accessibility to county-level hospitals was significant because of topography and transportation limitations. 

*Community-level healthcare accessibility*. The spatial accessibility of community-level health centers is shown in Figure 5c. One UCHI is in each urban community, and one THC is in each rural town; thus, accessibility should be equal at this level. In contrast, the CSA and ESA of community-level health centers covered only 13.98% and 58.62% of the total metropolis, respectively. There are many community-level HSAs scattered throughout the city. The community-level health centers clustered in the city center have the highest accessibility. The ESA in the eastern Longquan Mountain area is spatially fragmented. Moreover, the accessibility of community-level health centers is poor given the rugged topography of the Longmen Mountains. Residents in some settlements have to travel more than 2 hours to visit a THC. In particular, residents living in remote mountainous villages experience difficulties accessing primary healthcare because of unavailable transportation. The community-level health centers’ distribution is mostly scattered, while their accessibility is the worst among the three levels of healthcare services given the limited service capacity and residents’ distance tolerance.

*Multilevel healthcare accessibility*. According to the results (Figure 5d), Chengdu can be divided into the following eight types of healthcare access zones: the core service (CS) zone located in the CSAs of all levels of healthcare; the effective service (ES) zone located in areas outside the CS zone and inside the ESAs of all levels of healthcare; the lacking community-level healthcare (L-Com) zone; the lacking county-level healthcare (L-Cou) zone; the lacking city- and community-level healthcare (L-Cit&Com) zone; the lacking county- and community-level healthcare (L-Cou&Com) zone; the lacking city- and county-level healthcare (L-Cit&Cou) zone; and the lacking all levels of healthcare (LA) zone. The CS zone covers nearly all areas of the central city and is scattered in the suburban counties. The ES zone surrounds the CS zone and becomes fragmented on Longquan Mountain in the west and is cut off by Longmen Mountain in the east. In contrast to the CS and ES zones, the other six zones all lack certain levels of healthcare; thus, these zones are defined as multilevel healthcare shortage areas (MHSAs). The L-Com zone is the most widely distributed area and has a continuous distribution in the eastern and western mountainous areas and in Tianfu New District in the south and is scattered throughout the suburban areas. The L-Cou, L-Cit&Cou and L-Cit&Com zones are less widely distributed and are only scattered in sparsely settled mountainous areas. The L-Cou&Com and LA zones are clustered in the high mountainous areas on both sides of the city. 

#### 3.2.3. Multilevel Healthcare Spatial Equity

The criterion for multilevel healthcare spatial equity is whether the three levels of healthcare are distributed fairly across space according to the population demand. Therefore, this paper identified the proportion of the area and population covered by each multilevel healthcare access zone to evaluate the level of spatial equity (Figure 6). 

*Proportion of the area covered by each multilevel healthcare access zone* (Figure 6d). The results demonstrate that the ES zone covers 45.1% of the total area of the city, i.e., almost one-half of the metropolis is in the effective service scope of the three levels of healthcare. The residents in the CS zone, which covers 11.6% of the total area, can easily access all levels of healthcare. These two zones are recognized as adequate multilevel healthcare areas, while the remaining 43.3% are MHSAs. In the MHSAs, the most serious deficiency is the lack of primary care. A shortage in primary healthcare exists in the L-Com, L-Cit&Com, L-Cou&Com and LA zones, which cover 41.4% of the total area. Thus, 2/5 of the area of the city constitutes a shortage area that lacks community-level health centers. More importantly, 30.7% of the 41.4% of the total area has adequate city-level and county-level hospitals; nevertheless, the lack of primary care leads to an incomplete healthcare system that cannot meet the varying medical needs of the residents. In contrast, the L-Cou and L-Cit zones are relatively small and occur only in some remote mountainous areas. The LA zone covers only 3.2% of the city and is mainly distributed in high altitude mountain settlements with a low population density.

*Proportion of the population covered by each multilevel healthcare access zone* (Figure 6d). To evaluate healthcare access based on a combination of supply and demand, the population distribution in each healthcare access zone was further calculated. The ratio of the population settled in the ES zone is 46.0%, which is the highest of all the healthcare access zones. Although the CS zone only covers 11.6% of the area of the city, 43.3% of the total population lives in these areas. Therefore, 89.3% of the total population is in the effective healthcare service scope for all three levels. The remaining 11.7% of the population is settled in the multilevel healthcare shortage zones, which constitute 42.5% of the total area. Thus, 1.8 million people cannot visit all levels of healthcare within their travel distance tolerance. Moreover, 12.6 thousand people have to travel farther than an ordinary acceptable distance to visit any level of medical institution. Importantly, the majority of the population is settled in effective multilevel healthcare service areas, whereas some residents still have poor access to multilevel healthcare.

Although most of the healthcare shortage population is located in sparsely settled areas, we hoped to determine whether there were closely settled areas within MHSAs. Hence, maps of the gridded population and the multilevel healthcare access zones were overlapped to obtain the population distribution map of the multilevel healthcare access zones (Figure 6). The results show that in the MHSAs, the area north of Tianfu New District and the boundaries between Shuangliu County, Xindu District and the central city were significant closely settled areas with a high population density of more than 8000 people/km^2^. In addition, most suburban areas around the city center have a shortage of certain levels of healthcare, and the population density is 300–700 people/km^2^. These types of areas are fragmented and distributed at the administrative boundaries, with poor transportation. Meanwhile, public medical institutions are always located in the centers of the administrative districts, and residents are assumed to access services only within their region. Although residents who live in boundary settlements usually prefer to visit closer medical institutions than those in the administrative centers, the decision-making associated with allocating healthcare resources seldom considers residents that are not in the local jurisdiction. Regarding the sparsely settled mountainous areas (with a population density <50 people/km^2^) east and west of the metropolis where the healthcare shortage zones are continuously distributed, enhancing healthcare access could be extremely costly. Therefore, the reconstitution of scattered settlements into relatively clustered centers to enhance residents’ access to healthcare services is an operational countermeasure that is being implemented by local governments.

A shown in Figure 6c, the most closely settled areas located in the MHSAs only lack primary care, and a comparison with Figure 6d shows that 86.92% of the people among the 1.838 million living in the MHSAs only lack primary care. This result shows the significance of the spatial equity in primary healthcare for building a fair multilevel healthcare system in a metropolis.

## 4. Discussion

The results corroborate previous findings on the unfavorable geographic conditions for nonurban dwellers in accessing healthcare services [6,22]. In addition, several points need to be considered when interpreting the results of this study. First, higher-level healthcare has a lower quantity of providers and is more aggregated than lower-level healthcare; thus, evaluations of spatial equity using supply-driven methods might conclude that the higher levels of healthcare are allocated more unequally than the lower levels [28]. However, the results of this demand-driven study indicate that primary healthcare is allocated most unfairly based on the residents’ living patterns and travel distance preferences. Second, although the spatial inequity in access to multilevel healthcare evaluated based on the proportion of the population was relatively minor compared with that calculated based on the proportion of the area, a center-rich periphery-poor healthcare distribution pattern is also typical in this metropolis from the macro perspective. Moreover, this study identified many closely settled areas that are scattered in suburban and rural areas in MHSAs; therefore, arbitrarily defining different catchment sizes for rural and urban areas is not suitable for research on metropolises with large populations, varied topography, and rapidly changing urban spaces [13,29].

The main objective of the new healthcare reforms in China is to systematically optimize the spatial distribution of multilevel healthcare according to different resident demands [1,2,30,31]. The main findings of this study indicate that the equity of the multilevel healthcare service system should be improved through policies targeted at avoiding the short board effect. For instance, primary healthcare resources could be integrated and optimized across administrative regions at the boundaries of closely settled healthcare shortage areas; traffic accessibility could be improved in the suburban healthcare shortage areas to broaden the service scope of the existing medical institutions; and telemedicine and other support systems could be developed to promote high-quality medical resources in areas with larger urban agglomerations that extend to remote mountainous areas.

There are several limitations associated with this study that should also be noted. Because of data availability restrictions, we could not include the number of doctors and hospital beds in the calculations of healthcare accessibility, and such detailed data would be highly significant in evaluating healthcare access from a quality perspective. Furthermore, access to healthcare is multidimensional, and this study focused on spatial factors, but nonspatial factors, such as socioeconomic status, were not considered. Moreover, the diversification of healthcare service providers is a trend that is designed to meet the increasing needs for healthcare, but this study only assessed public healthcare institutions. 

Future research efforts could focus on applying these dynamic datasets to other scenarios. For example, night-light data could be upgraded for the latest population distribution; residents’ spatial preferences could be investigated under different cultural contexts and for different healthcare demands; and real-time traffic data could be calculated via different traffic choices and for different time periods to generate additional temporal pattern estimations. A variety of socioeconomic factors, including residents’ demographic characteristics, the built environment and other local features should be considered in future studies [32]. Within the metropolitan area there is an increasing number of private hospital and rehabilitation institutions providing different types and levels of healthcare services. Similarly, the medical service demands of residents is becoming diversified. Some special medical service demands cannot be met by the traditional public hospitals, which need the supplement of private medical institutions. However, the coordination and competition between public and private medical institutions is still unclear, and private medical institutions may be concentrated in urban centers due to market allocation, which would further aggravate healthcare inequity between urban and suburban areas. The integration of these diversified entities represents an essential direction for future research [33]. 

## 5. Conclusions

This study proposed the “2R grid-to-level” (2R-GTL) approach for performing systematic and dynamic assessments of the equity in the spatial accessibility to a multilevel healthcare system in a metropolis. Two main innovations resulted from this study. First, rather than identifying the accessibility from residential locations to separate hospitals, we provided a method of evaluating the spatial accessibility from each grid of a city to each level of healthcare, which has practical significance regarding the strategic and systematic planning of hierarchical healthcare systems in a metropolis. Second, we combined sophisticated geocomputational tools and an individualized survey through the 2R method to obtain more precise living patterns and spatial behavioral patterns of residents, which was highly important for accurately and dynamically identifying the healthcare shortage areas and populations from a geographic demand-driven perspective.

Based on this new method, three important findings can be drawn from this empirical study. First, there are significant differences in the spatial accessibility to different levels of healthcare in Chengdu. City-level hospitals have the highest accessibility, while community-level health centers have the lowest accessibility. Second, the lack of primary healthcare is the most serious problem in these healthcare shortage zones. In total, 10.17% of the total population and 86.92% of the population settled in the MHSAs are lacking only primary care. Third, the MHSAs are mainly clustered in the sparsely settled mountainous areas on the west and east sides of the metropolis, where the rugged topography and poor transportation are major access barriers. Meanwhile, some closely settled areas that lack multilevel healthcare are scattered in the suburban administrative boundary areas.

## Figures and Tables

**Figure 1 ijerph-16-00493-f001:**
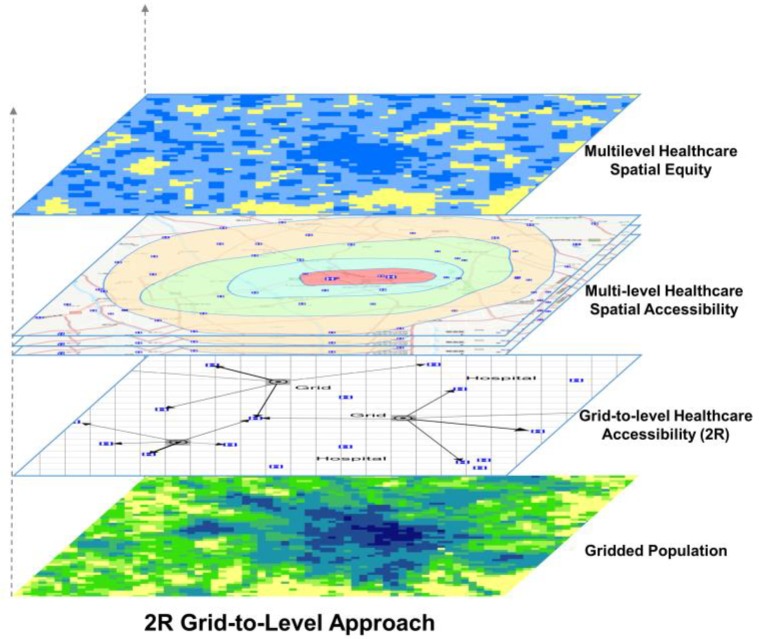
2R Grid-to-level approach.

**Figure 2 ijerph-16-00493-f002:**
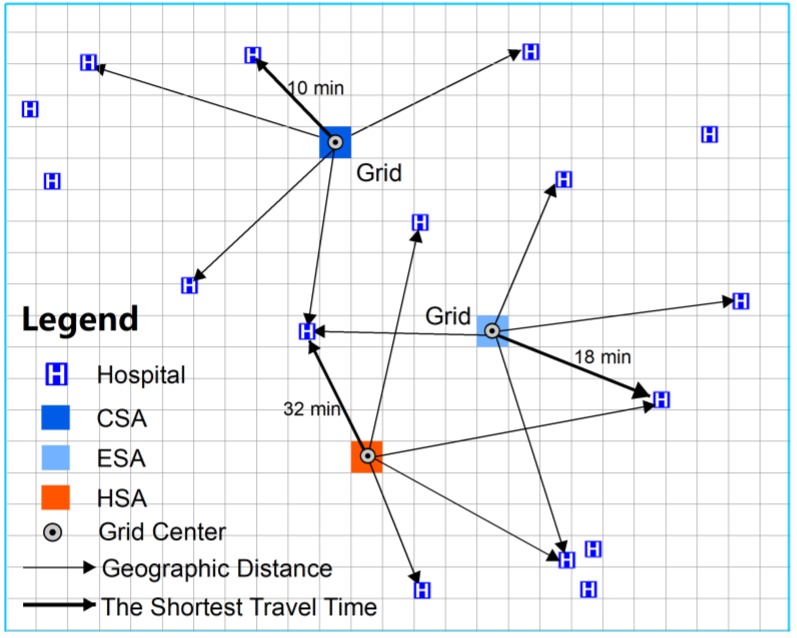
Calculation of grid-to-level healthcare spatial accessibility.

**Figure 3 ijerph-16-00493-f003:**
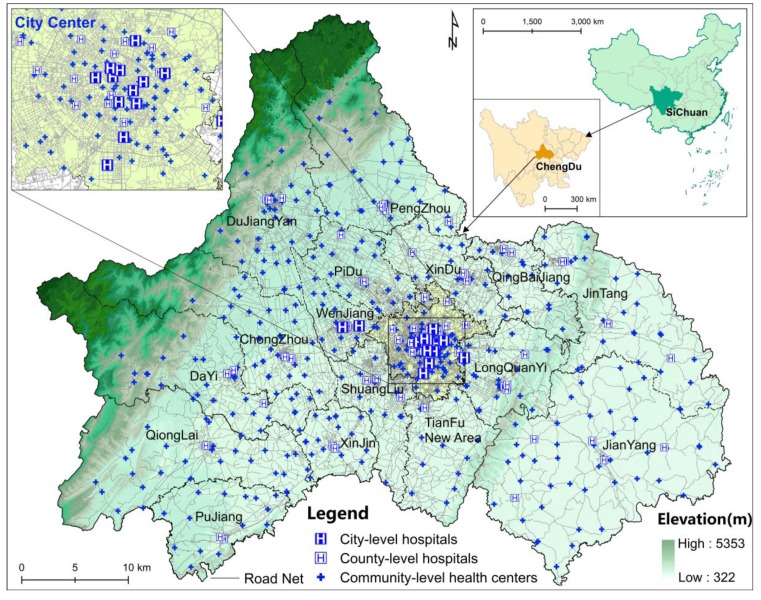
Overview of Chengdu and distribution of healthcare institutions.

**Figure 4 ijerph-16-00493-f004:**
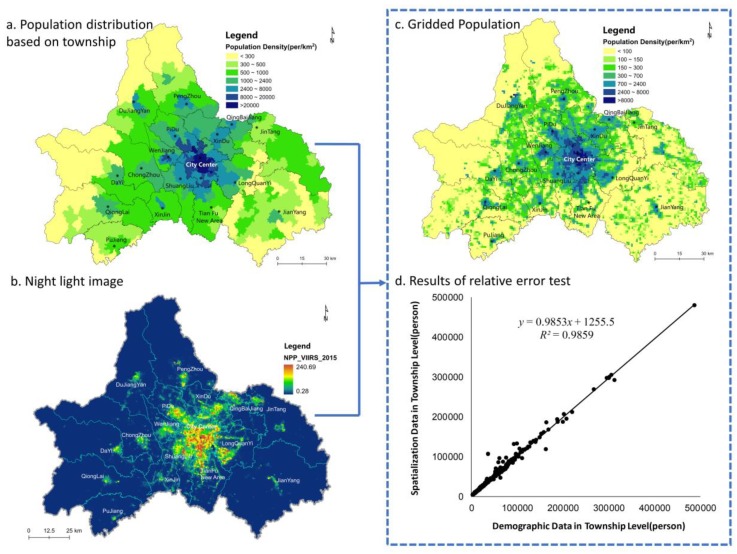
Processing of population gridding.

**Figure 5 ijerph-16-00493-f005:**
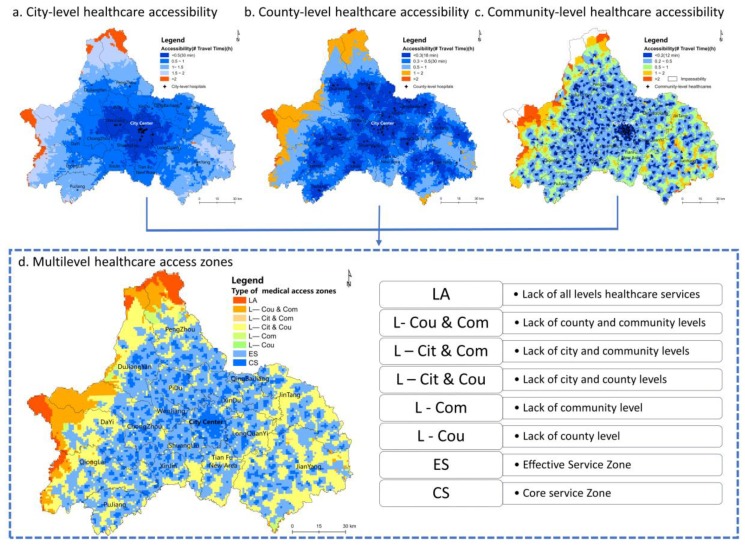
Multilevel healthcare accessibility.

**Figure 6 ijerph-16-00493-f006:**
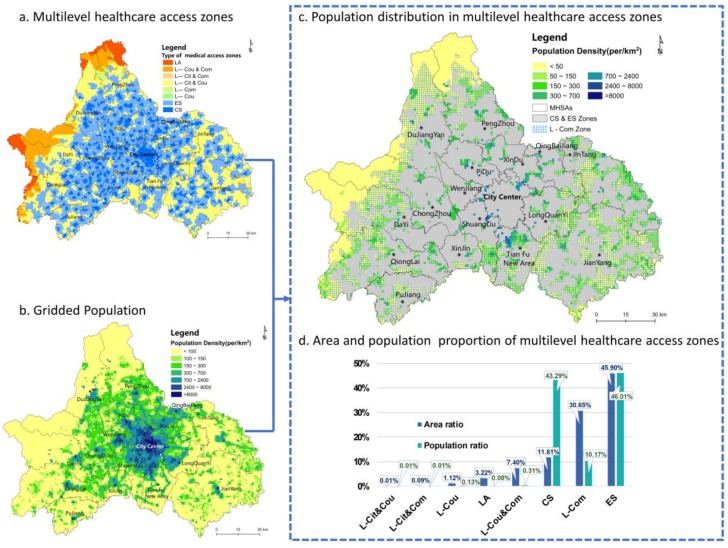
Multilevel healthcare spatial equity.

**Table 1 ijerph-16-00493-t001:** Multilevel healthcare system and function of each level.

Level	Definition	Examples	Function
city-level hospitals	large provincial and metropolitan general hospitals	the People’s Hospital of Sichuan Province	Concentrate on serious illnesses and complex diseases
county-level hospitals	hospitals belonging to city districts and counties	the Peoples Hospital of Xindu District	Focus on common diseases and frequently occurring illnesses
community-level health centers	urban community health institutions (UCHIs) town health centers (THCs)	the UCHI in Longteng community the THC in Majia town	Perform the first diagnosis and rehabilitation therapy

**Table 2 ijerph-16-00493-t002:** Residents’ spatial behavioral preferences of visiting multilevel healthcare.

Residents’ preferences	Types of access areas	City-level hospital	County-level hospital	Community-level health centers
Maximum travel time	Effective service area (ESA)	2 hours	1 hour	0.5 hour
Ordinary acceptable travel time	Core service area (CSA)	1 hour	0.5 hour	0.2 hour
Traffic Choice		Driving car	Driving car	cycling and walking

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
