# Peer review of "Spatial Equity of Multilevel Healthcare in the Metropolis of Chengdu, China: A New Assessment Approach"

_ijerph, 2019, doi:10.3390/ijerph16030493_

Round 1
Reviewer 1 Report
Figures order ok
Table order ok
Figure resolution ok
Logical sequence of text ok
- Interesting article regarding societies organization. A location in China was studied. A method was proposed. New results were obtained making possible for decision-makers a careful planning of the distance among public health facilities, healthcare, transportation and individuals behavioral patterns.
- Square kilometer, check notation at lines 112, 125, 141, 348, 350, 356.
- According to template, “In the text, reference numbers should be placed in square brackets [ ], and placed before the punctuation; for example [1], [1–3] or [1,3].”, it is necessary to make a difference if reference start from 1 to 3, including 2, or it is only 1 and 3. Check references in brackets. For example, at lines 73 and 77, it was found the same notation. They should be corrected all over the article.
- Page 3, 6, 9, 10 and 11 is without any citation, is that correct? Just an observation for the authors to address the reviewers.
- Line 104 – 106, where it says section 4 drawn the main highlight conclusions. Check if it is located in section 5, not mentioned in those lines. Or if section 4 is as stated later in text, the discussion part. This is just an observation. Authors should clear the reviewer doubt.
- Some references are without DOI. According to template they should be added at reference list. For example, reference 18, DOI is available: https://doi.org/10.1016/j.jth.2017.03.013.
Check other references for DOI.
Author Response
Response to Reviewer 1 Comments
Point 1 Square kilometer, check notation at lines 112, 125, 141, 348, 350, 356.
Response 1: Thanks, we have checked the full text for similar errors and made corrections (Yellow Highlight).
Point 2 According to template, “In the text, reference numbers should be placed in square brackets [ ], and placed before the punctuation; for example [1], [1–3] or [1,3].”, it is necessary to make a difference if reference start from 1 to 3, including 2, or it is only 1 and 3. Check references in brackets. For example, at lines 73 and 77, it was found the same notation. They should be corrected all over the article.
Response 2: Thanks for your suggestions. In this version, we carefully distinguished the two types of citation and made corresponding modifications (Yellow Highlight).
Point 3 Page 3, 6, 9, 10 and 11 is without any citation, is that correct? Just an observation for the authors to address the reviewers.
Response 3: Thanks for your comments. These pages are mainly the overview of the research area, the introduction of data, methods and research results, it is mainly original content, so there is no citation. However, in the introduction and discussion, the existing relevant literature is fully cited.
Point 4 Line 104 – 106, where it says section 4 drawn the main highlight conclusions. Check if it is located in section 5, not mentioned in those lines. Or if section 4 is as stated later in text, the discussion part. This is just an observation. Authors should clear the reviewer doubt.
Response 3: Thanks for your comment. We note the inappropriateness and inaccuracy of the statement in the text and have amended it. Section 4 make a comprehensive discussion of the results and highlights the main conclusions drawn from this study in Section 5.
Point 5 Some references are without DOI. According to template they should be added at reference list. For example, reference 18, DOI is available: https://doi.org/10.1016/j.jth.2017.03.013.
Check other references for DOI.
Response 3: Thank you for your suggestion. We have marked DOI for all references.

Reviewer 2 Report
I think the topic of spatial equity of healthcare systems is very interesting. I think an idea to extend the paper would be to include prices of healthcare services and median income of residents in the zones, specifically in a behavioral model to estimate quantity of services demanded at each level.
Author Response
Response to Reviewer 2 Comments
Point 1 I think the topic of spatial equity of healthcare systems is very interesting. I think an idea to extend the paper would be to include prices of healthcare services and median income of residents in the zones, specifically in a behavioral model to estimate quantity of services demanded at each level.
Response 1: We are very appreciative for your valuable suggestion which is an exciting and challenging research idea and a great motivator for us. Following your comment, we plan to further extend the research to individual level and integrate residents’ behavior of selecting medical treatment and the accessibility model. Prices of healthcare services, time spent on medical treatment, health status, income and occupation, etc. would be included in our future research.

Reviewer 3 Report
This is an exceptionally well-written manuscript, and one that adds to the literature. I have only a few point to raise.
Not sure that I agree with the first sentence on main reasons for disordered medical care…there are likely to be many, and so insert a reference here to justify. I found a few similar type judgement statements, and so it might be worth a good review.
The level of complexity (and the number of figures) is large. If there were any way in which to simplify the results (for the readership that is not familiar with the methods or the technology), it would be welcomed.
Private settings were not included. Some further speculation as to how these data points would/will alter the results should be further elaborated upon in the discussion or limitations.
It there any reason to believe that aligning specialist services (even primary care) according to specific population health demand, could alter your results. Geographical access to care does not mean that people are receiving what they actually need in terms of care.
Author Response
Response to Reviewer 3 Comments
Point 1 Not sure that I agree with the first sentence on main reasons for disordered medical care…there are likely to be many, and so insert a reference here to justify. I found a few similar type judgement statements, and so it might be worth a good review.
Response 1: We sincerely thank you for pointing out this problem. The unproved judgment is indeed somewhat arbitrary, thus, we have modified the sentence and inserted references per your suggestion. A detailed explanation of this judgement statements can be found in lines 84-87.
Point 2 The level of complexity (and the number of figures) is large. If there were any way in which to simplify the results (for the readership that is not familiar with the methods or the technology), it would be welcomed.
Response 2: Thank you for your suggestion. We proposed a new approach to estimate the multilevel healthcare accessibilities which could be used in other empirical studies. Thus, we explained our research methods and results in details. The calculation process of this paper is mainly divided into population gridding and accessibility computing. The detailed calculation formula of population gridding can be found in the paper. The accessibility computing is mainly completed by Python coding, and we have submitted the source code for the readers who would like to use this method but not familiar with this technology. The figures in this paper were mainly made based on ArcGIS software, and the relationship between different figures is demonstrated by composed figures to help our readers have better understanding of the logic of our research results.
Point 3 Private settings were not included. Some further speculation as to how these data points would/will alter the results should be further elaborated upon in the discussion or limitations.
Response 3: Thank you for your important suggestions. It is a pity for failing to include private medical institutions in this paper for data limitation. Following your opinion, we have made modifications in the discussion to elaborate on the importance of private medical institutions to the healthcare equity. Within the metropolitan area there is an increasing number of private hospital and rehabilitation institutions providing different types and levels of healthcare services. Similarly, the medical service demands of residents is becoming diversified. Some special medical service demands cannot be met by the traditional public hospitals, which need the supplement of private medical institutions. However, the coordination and competition between public and private medical institutions is still unclear, and private medical institutions may be concentrated in urban centers due to market allocation, which would further aggravate healthcare inequity between urban and suburban areas. Therefore, we will try other methods to collect data and our future research would include private medical institutions in the accessibility model per your suggestion.
Point 4 It there any reason to believe that aligning specialist services (even primary care) according to specific population health demand, could alter your results. Geographical access to care does not mean that people are receiving what they actually need in terms of care.
Response 4: Thank you for your comments. In our results, there are indeed some densely populated areas in the MHSAs district, whose medical service needs should not be ignored (Figure 6). And more, 10.17% of the total population and 86.92% of the population settled in the MHSAs are lacking only primary care. In the government's healthcare planning, it is possible to mitigate this inequality, especially in primary care. For example, new community health service centers could be planned in some densely populated communities. Indeed, geographical access to care does not mean that people are receiving what they actually need in terms of care, but geographical accessibility is the basis of practical accessibility. It is not perfect to only consider the geographical accessibility, other factors that affect the actual accessibility of residents should be included in the accessibility model (in our discussion). However, it is difficult to complete a perfect accessibility evaluation in single article. Following your comment, we will collect more data, particularly focus on nonspatial data such as economic status, healthcare insurance, time and professional availability, and design more accurate evaluation models to constantly promote the evaluation of medical equity.
